# Vinylated Modification of Biophytic Acid and Flame-Retardant/Crease-Proofing Finishing of Cotton Fabrics via In Situ Copolymerization

**DOI:** 10.3390/ma16010286

**Published:** 2022-12-28

**Authors:** Bingying Cheng, Qingqing Zhou, Jiayi Chen, Xu Zhang, Chenglei Zhu, Minghao Wu

**Affiliations:** College of Textile and Clothing, Yancheng Institute of Technology, Yancheng 224051, China

**Keywords:** flame-retardant and crease-proofing finishing, phytic acid, cotton textiles

## Abstract

The vinyl phytic acid (GPA) was prepared using biophytic acid (PA) and glycidyl methacrylate (GMA), in which double bonds were introduced into the phytic acid molecule to increase the active groups in the phytic acid molecule. Furthermore, itaconic acid (IA) containing two unsaturated double bonds and GPA was polymerized in situ and crosslinked on the surface of cotton fabrics, and flame retardant and crease-proofed fabrics were obtained. The effects of GPA, IA, and the initiator on the flame-retardant and crease-proofing properties of the fabrics were analyzed by a single-factor and double-dip double-nip experiment. A flame-retardant and wrinkle-resistant fabric was obtained when the limiting oxygen index (LOI) and wrinkle recovery angle (WRA) were 28% and 270°, respectively. During combustion, the thermal properties of the fabrics changed; typically, the extrapolated initial temperature (Te) decreased, and moisture release increased. After burning, the fabrics had good shape retention, and the carbon residue content increased to 48%, which effectively inhibited or slowed down the combustion and heat release of the textiles. However, the whiteness, mechanical properties, and washability of the products need to be further improved.

## 1. Introduction

As one of the most favorable natural fiber fabrics, cotton is widely applied in all kinds of clothing, home, and industrial textiles because of its good comfort, moisture absorption, and permeability. Nevertheless, the LOI of cotton fabric is only 18%, which is a flammable fiber. When encountering an open fire, it burns quickly, which has contributed to restricting its application. Research into the flame-retardant finishing of cotton textiles has become the focus at home and abroad to restrain fabric burning.

Traditional flame retardants for cotton may cause some problems, such as toxicity, low efficiency, no durability, and harm from formaldehyde release, which have attracted more and more attention. At present, the durable and low-toxicity flame retardants for cotton that are widely used in the market include n-hydroxymethyl-3-dimethoxyphosphoryl propionamide (pyrovatex CP) and tetrahydroxymethyl phosphorus chloride (THPC) and its improved products. However, there is a problem with releasing formaldehyde [1]. Formaldehyde is regarded as a carcinogenic source by the World Health Organization [2]. Therefore, a new development direction for flame retardants for cotton is to seek new environmental flame retardants with high efficiency, excellent durability, low smoke, low toxicity or no toxicity, and no formaldehyde. Some scholars had prepared a new type of dendriform poly(amine) dendrimer flame retardant, which was used for the flame-retardant finishing of cotton fabric under the action of citric acid. The limiting oxygen index exceeded 25%, and the carbonization length of the treated sample after washing five times was slightly longer than that of the unwashed sample [3].

Phytic acid (PA), with a molecular formula of C_6_H_12_O_24_P_6_, a relative molecular weight of 660.08, and a phosphorus content of 28.16%, is widely found in cereal, beans, vegetables, bran, rice bran, and other plant seeds [4]. PA has been gradually used in the flame-retardant finishing of textiles because of its relatively high phosphorus content and wide sources. At present, PA flame-retardant finishing has been studied more in protein fiber and composite materials, mainly because protein fiber molecules contain more active groups, such as amino and carboxyl groups, which have the ability to directly form ionic bonds with phytic acid or form amino metal ion–phytic acid ternary complexes. After 30 washing cycles, the fabric still had a self-extinguishing effect [5,6,7]. The flame-retardant finishing of composites usually uses insoluble phytate substances (such as nickel phytate, barium phytate, etc.) to impart flame-retardant properties to composites by means of blending or film formation [8,9]. It has also been reported that phytic acid is used for flame-retardant finishing of cellulose fibers. However, because the macromolecules of cellulose fibers contain hydroxyl groups, oxygen anions are formed in neutral or alkaline solutions, which have the same charge as phytic acid anions, and there is electrostatic repulsion. Moreover, the complexation ability of hydroxyl groups on cellulose molecules is not as good as that of amino and carboxyl groups on protein fibers, so the flame-retardant finishing of phytic acid is relatively difficult. The existing literature reports that the flame-retardant finishing of cotton fabric by phytic acid basically adopts a layer-by-layer assembly method [10,11], and the layer-by-layer assembly involves the alternating deposition of molecules with positive and negative charges to build multilayer films on a substrate [12,13]. Furthermore, researchers from Southwest University [14] used urea to modify phytic acid to prepare ammonium phytate and then used dicyandiamide as a catalyst to finish cotton fabric. The LOI of the flame-retardant fabric was 43.2%, and after washing it 30 times, the LOI was 30.5%. Compared with the untreated cotton fabric, the peak heat release rate and total heat release rate of the treated cotton fabric decreased significantly. Subsequently, based on ammonium phytate, the researchers [15] synthesized raffinose crosslinked boric acid–ammonium phytate (r-GBAP) flame retardant by the one pot method to enhance the durable flame retardancy of cotton fabrics. When the amount of flame retardant (r-GBAP) was 100 g/L, the LOI of the flame-retardant fabric was 41.6%, which has water washing resistance. In addition, Li [16] selected ammonium phytate and chitosan as environmentally friendly flame retardants to improve the flame retardancy and antibacterial properties of viscose fabrics. The thermal stability of flame-retardant viscose fabric in a higher temperature zone was significantly improved. The limiting oxygen index of the viscose layer-by-layer assembly of the two layers was 29%, and 39.0% of the coke residue was retained at 700 °C. However, the flame-retardant and crease-resistant finishing of cotton fabric with phytic acid has not been reported.

The work in this chapter is a new way. Glycidyl methacrylate (GMA) was used to modify phytic acid, and vinyl phytic acid (GPA) was obtained. Furthermore, itaconic acid (IA) containing two unsaturated double bonds and GPA were polymerized in situ and crosslinked on the surface of cotton fabrics. Flame-retardant and crease-proofed fabrics were obtained. Flame retardancy was tested using LOI. Crease performance was evaluated by WRA. The application of scanning electron microscopy (SEM), Fourier transform infrared spectroscopy (FTIR), and X-ray photoelectron spectroscopy (XPS) was mainly to investigate the morphology and surface composition of the treated and untreated fabrics. Furthermore, changes in thermal stability and combustion behavior were assessed by thermogravimetric analysis (TG) and thermogravimetry–Fourier transform infrared spectrometry (TG-FTIR) in a nitrogen atmosphere.

## 2. Experimental

### 2.1. Materials

Bleached twill cotton fabric (weight: 130 g/m^2^) was purchased from a local market. Acetone (AC), methanol (MeOH), and sodium hydroxide (NaOH) were analytically pure and supplied by Sinopharm Chemical Reagent Co., Ltd. Glycidyl methacrylate (GMA), itaconic acid (IA), phytic acid (PA, 70 wt.%), sodium hypophosphite (SHPP), ammonium persulfate (APS), and butane tetracarboxylic acid were also analytically pure and provided by Aladdin Reagent Co., Ltd. (Shanghai, China).

### 2.2. Preparation of GPA

The vinyl modification of phytic acid (GPA) was carried out according to the preparation methods of reference [17]. GPA was synthesized via the phosphoryl group of PA and the epoxy group of GMA by open-ring esterification. The specific methods are as follows: The mixture of 13.2 g PA and 80 mL AC was put into necked flask. A total of 8.5 g glycidyl methacrylate and 0.24 g p-hydroxy anisole inhibitor were mixed and added dropwise into necked flask containing PA acetone solution within 1 h. Subsequently, the above solution was stirred continuously for 5 h at 65 °C. At the end of the reaction, a brown solution was obtained, and then the acetone solvent was removed by vacuum distillation, and unreacted GMA was removed by toluene. Eventually, white powder was prepared by precipitation with 40 g/L sodium methoxide solution, filtered, washed with methanol three times, and vacuum dried at 50 °C for 24 h and weighed. The conversions were all about 72% according to the mass change before and after the reaction. The powder was marked as GPA, ^1^H NMR, as shown in Table 1.

### 2.3. Flame-Retardant and Crease-Proofing Finishing of Cotton Fabrics

The bleached cotton fabrics were treated at 90 °C for 20 min in 2 g/L NaOH solution with bath ratio of 1:30, washed twice in hot water at 80 °C, soaked in 1 mL/L acetic acid solution for 5 min at room temperature, and washed with cold water for standby.

In the presence of 2% butane tetracarboxylic acid, the effects of different dosages of GPA (10%, 15%, 20%, 25%, 30%), IA (mass ratio of IA to GPA of 1.5:1, 1:1, 1:1.5, 1:2, 1:2.5), APS (mass fraction of monomer of 6%, 8%, 10%, 12%, 14%), and SHPP (mass ratio of IA to SHPP of 2:1, 1.5:1, 1:1, 1:1.5, 1:2) on the flame retardancy, crease performance, and whiteness of the fabric were investigated. The fabrics were dipped and rolled twice at 100% liquid ratio. The sample was prebaked at 100 °C for 5 min and baked at 180 °C for 90 s. The original fabric was marked as O-C, and the finished fabric was as GPA-C.

### 2.4. Characterization

Limiting oxygen index (LOI) was calculated according to the “GB/T 5454-1997 Textile Combustion Performance Test Oxygen Index Method”, and the flame retardancy of cotton fabric was studied by using oxygen index tester (HC-2, China). Fabrics (10 cm × 5 cm) were prepared and placed in a cylinder, where a mixture of N_2_ and O_2_ was passed. The ratio of N_2_ and O_2_ was adjusted to determine the limiting oxygen index (LOI) of the sample.

The whiteness (Wg) of the fabrics was tested on WSD-III automatic whiteness tester in accordance with the “GB/T 8424.2-2001 Textile Color Fastness Test Method for Instrumental Evaluation of Relative Whiteness.”

Breaking strength (BS) was determined using 30 cm × 5 cm samples according to “GB/T 3923.1-2013 Textile Fabric Tensile Properties Part 1: Determination of Breaking Strength and Elongation at Break (Strip Method)”. The longitudinal breaking strength of the fabrics was measured by YG065 electronic fabric strength machine (tensile).

Tear strength (TS) was evaluated according to “GB/T 3917.1-2009 Textile Tear Performance of Fabrics Part 1: Determination of Tear Strength by Impact Pendulum Method”. Samples were prepared according to requirements, and the tear strength of fabrics was determined using YG (B)033A fabric tearing apparatus.

Wrinkle recovery angle (WRA) was assessed according to “AATCC 66-2008 Fabric Wrinkle Recovery: Recovery Angle Method”, in which five warp and five weft samples were prepared and measured by YG (B) 541D-II automatic digital fabric wrinkle elastic meter.

The crystal region of the samples was measured by X-ray diffraction (XRD) multifunctional powder (Netherlands X’Pert3 powder). The voltage of the tube was 36 kV, and the current was 20 mA. The scanning range (2*θ*) was 10–70°, and the scanning step was 0.02° in *θ*/2*θ* mode.

SEM images were taken using a Nova Nano SEM 450 scanning electron microscope (FEI, Hillsborough, OR, USA). The surfaces of the fabrics were plated with gold before observation. Elemental analysis (EDX) of the samples was conducted using Aztec X-MAX 80 multifunctional surface electron spectroscopy (Oxford Instruments, Oxford, UK).

With a thermos X-ray source system (ESCALAB 250XI, Waltham, MA, USA), X-ray photoelectron spectroscopy (XPS) analysis was performed at a reduced power of 100 W and a mono Al Ka X-ray source (1486.6 eV).

## 3. Results and Discussion

### 3.1. ATR-FTIR Analysis of GPA

Figure 1 shows the infrared spectra of PA and GPA. In the PA curve, 3300 cm^−1^, 1200 cm^−1^, and 1070 cm^−1^ respectively correspond to the vibration characteristic absorption peaks of the -OH, P=O, and P-O groups in PA [18]. By comparison with the infrared spectrum curve of PA, there are new characteristic absorption peaks near 1720 cm^−1^, 1470 cm^−1^, 1380 cm^−1^, and 1002 cm^−1^, which correspond separately to the bending vibration of C=O, CH, the bending vibration of CH_3_, and the stretching vibration of C=O-O in the infrared spectrum curve of the GPA product. The characteristic peaks are basically in line with that of the groups in the GMA molecule. At the same time, this shows no difference between the absorption peak of the C=C group near 1640 cm^−1^ in the product and the C-H stretching vibration in the PA, while there is no characteristic absorption peak near 815 cm^−1^ or 908 cm^−1^ attributed to the epoxy skeleton [19], which indicates that the epoxy group opens the ring under strongly acidic conditions.

### 3.2. Preparation of GPA and Graft Reaction Principle of Cotton Fabrics

Maleic acid (MA), itaconic acid (IA), and other unsaturated carboxylic acid molecules contain two carboxyl groups and C=C double bonds, which can be used for the antiwrinkle finishing of cotton fabrics. Because of its molecular structure, itaconic acid is more prone to free radical polymerization than maleic acid [20]. Therefore, itaconic acid and modified phytic acid were used to finish the fabrics. Figure 1 was the reaction mechanism of PA and GMA, and Figure 2 was the graft reaction principle of cotton fabrics. In the process of high-temperature baking, the products in Figure 2 may be formed between the same or different molecules. Among them, reaction products (1) and (3) are beneficial to the flame-retardant and wrinkle-resistant effect of the fabric. Under the action of the catalyst, the double bond in the GPA structure and the double bond in the IA structure are polymerized first, and then the carboxyl group in the itaconic acid and the hydroxyl group of the cotton fabric are esterified, and the phytic acid and itaconic acid are successfully grafted onto the cotton fabric, which is beneficial to the improvement in the flame retardancy and wrinkle resistance of the fabric. In Reaction (2), GPA is not polymerized with itaconic acid and is only deposited on the surface of the fiber. During the washing process, the phytic acid is easy to fall off. Therefore, Reaction (2) is only effective for the wrinkle resistance of the fabric. During the finishing process, the steric hindrance effect in the molecular structure of the GPA has a certain effect on the formation of Reactions (1) and (3), so the occurrence of Reaction (2) is inevitable.

### 3.3. Optimization of Finishing Process

#### 3.3.1. Effect of IA: GPA on the Cotton Fabrics

In the IA molecule, IA contains two carboxyl groups and C=C double bonds. The double bonds and GPA are polymerized in situ, and then the carboxyl group can be crosslinked with cotton fabric. PA is grafted indirectly onto the surface of cotton fabrics to improve the flame retardancy and wrinkle resistance of the fabrics. The effect of an IA dosage on fabric properties was determined under the following conditions: PA concentration of 15%, IA: SHPP molar ratio of 1:1, and APS to monomer mass fraction of 10% (Figure 2). In the presence of a small amount of IA (<7.5%), the LOI and whiteness of the fabrics were lower than 24% and 71%, respectively, which may be due to the failure of GPA to polymerize fully with IA. During curing, the fabrics were yellow because of the unreacted double bond. Moreover, the crosslinking degree with IA was low, resulting in a decline in the flame retardancy and whiteness of the fabric. As the IA increased, the small molecule IA reduced the steric hindrance effect of GPA to a certain extent, which increased the degree of in situ copolymerization between GPA and IA and improved the flame retardancy and whiteness of the fabric. With increasing IA concentration, the carboxyl group in the structure of IA was grafted onto cotton fabric, thereby improving the crease resistance. However, the mass ratio of IA to GPA was 1:1 because IA was not easy to dissolve in high concentrations and at room temperature.

#### 3.3.2. Effect of IA: SHPP on the Cotton Fabrics

The effect of SHPP on fabric properties was studied by changing the ratio of IA: SHPP under the following conditions: GPA concentration of 15%, IA: GPA mass ratio of 1:1, and APS on monomer mass fraction of 10%. With an increasing ratio of IA:SHPP to 1:1.5, the flame-retardant (23%→27%) and crease-resistant properties (245°→270°) of the fabric were improved. As the concentration of SHPP was continuously increased, the flame-retardant and crease properties of the fabric increased slowly. The increase in SHPP, a catalyst of esterification, was found to be beneficial to the reaction of the carboxyl group in the BTCA and IA structure with the hydroxyl group on fabrics. Meanwhile, SHPP formed an oxidation–reduction system with the oxidant APS, reduced the temperature of polymerization, and improved the crosslinking degree of GPA and IA, thereby enhancing the flame retardancy and crease resistance of fabrics. The finishing solution was strongly acid, and SHPP was alkaline. The pH of the solution increased with SHPP, and the whiteness of the fabric was perfect. However, a specific white solid was deposited on the surface of the dried fabric due to the low solubility of SHPP at room temperature. The deposited solid easily turned yellow after high-temperature baking, and the whiteness of the fabrics decreased. This finding is basically consistent with that in the existing literature [20]. Considering the flame retardancy, crease resistance, and whiteness of the fabrics, the ratio of IA: SHPP was set as 1:1.5.

#### 3.3.3. Effect of APS on the Cotton Fabrics

APS, as a catalyst for in situ polymerization, determines the degree of polymerization of GPA and IA and affects the flame retardancy of fabrics. Figure 3A shows the effect of APS dosage on fabric properties. The effective polymerization degree of GPA and IA increased with APS monomer content, and a polymer film was formed on the surface of the fabric. The carboxyl group in IA and the hydroxyl group in the cotton fabric reacted, resulting in improved flame retardant and wrinkle properties of the fabric. As an oxidant, APS oxidized the fabric during high-temperature baking. At higher concentrations, the oxidation was stronger, and the loss of the whiteness of the fabric was greater. APS and SHPP formed an oxidation–reduction system, which reduced the catalytic activity of SHPP in the esterification of carboxyl and hydroxyl groups to a certain extent, thereby decreasing the crease-resistant property of the fabric. After comprehensive analysis, APS was selected to be 10% of the monomer dosage.

#### 3.3.4. Effect of GPA on the Cotton Fabrics

PA, which contains 6 P and 28.16%, is an organic phosphorus additive that is characterized by abundant raw materials, environment friendliness, and good biocompatibility [21]. GPA obtained by esterification with GMA and PA was more conducive to application as a flame retardant. The effect of the GPA content on fabric properties was discussed under the following conditions: mass ratio of IA: GPA of 1:1, mole ratio of IA: SHPP of 1:1.5, and mass fraction of APS to monomer of 10%. The flame retardancy of the fabric increased from 25% to 30% with increasing GPA content. However, the crease-resistant property of the fabric decreased when the concentration of GPA was decreased to 20%, which may be due to the fact that the presence of hydroxyl group in the GPA molecule hindered the esterification reaction between the carboxyl group in IA or BTCA and the hydroxyl group in cotton fiber. The whiteness of the fabric showed a downward trend due to the GPA brown solution. Hence, the GPA concentration for the successful test was selected as 20%.

In conclusion, the optimal process conditions of fabric finishing were as follows: bath ratio of 1:30, mass ratio of IA: GPA of 1:1, mole ratio of IA: SHPP of 1:1.5, and GPA concentration of 20%. In situ polymerization and crosslinking finishing occurred on the surface of the cotton fabric under an APS to monomer mass fraction of 10% to improve the flame retardancy and wrinkle resistance of the fabrics.

### 3.4. Fabric Performance

#### 3.4.1. Basic Properties of Finished Fabrics

Experiments were carried out under optimized conditions to test the basic properties of fabrics (Table 2). The LOI and WRA of the finished fabric increased compared with those of the original fabric; however, the whiteness and mechanical properties were seriously damaged, especially the tear strength. The LOI of the finishing fabric reached 28.6, and the WRA was increased to 265°; both parameters decreased after washing because a number of IA did not react with the hydroxyl group of the fabric. In the presence of the catalyst, the hydroxyl group in the molecular structure of PA blocked the grafting reaction of IA and the fabric. The steric hindrance of PA also reduced the polymerization with IA. Therefore, the LOI and WRA of the fabrics decreased to a certain extent. In addition, the whiteness and mechanical properties of the fabric were seriously reduced because fabric finishing was carried out under an acidic solution with the oxidant. In situ polymerization and crosslinking should be further studied to improve the wearability and added value of the fabric.

#### 3.4.2. SEM and EDX Analysis

The apparent morphology (5000 times) and energy spectrum before and after finishing and burning are shown in Figure 4. The surface of the original fabric (A1) was smooth and in a state of ash, shriveling, and curling after burning. A film was formed on the surface of the finished fabric (A2). The burned fiber was in good shape, and the presence of bubbles on the surface of the fabric remained a wonder. The reason may be that IA had low solubility at room temperature, and its sodium salt easily precipitated in the finishing solution. During drying, the solid deposited on the surface of the fabric, and bubbles appeared when burned. Based on the energy spectrum, P and Na exist in the fabric before and after combustion, consistent with the XPS results.

#### 3.4.3. XPS Analysis

Figure 5A shows the XPS spectrum of the fabrics before and after finishing. The contents of C and O in the fabrics were high, and the finished fabrics contained not only C and O but also P and Na. Figure 5B–D shows the existing forms of P, O, and C on the surface of finished fabrics. P existed in two forms: an ester bond (C-O-P) at about 133.1 eV and a PO_4_^3−^ group at 132 eV [22,23,24]. Cellulose carbon atoms included three binding forms, namely, C1, C2, and C3 at 284.7, 286.4, and 288.6 eV in Figure 5D (C1s), respectively [25]. C1 was low electron-binding energy (284.7 eV) and belonged to C-H or C-C; C2 was connected to an enormous number of -OH groups in the fiber, and the electron-binding energy was increased to 286.4 eV due to the high electronegativity of the -OH groups. C3 was the C-O-C in the fiber, and its peak position was about 288.6 eV. The energy spectrum peaks at 530.7 eV, 532.7 eV, and 531.5 eV belonged to two modalities of the oxygen atom, namely, =O- (P=O—or C=O-), C-O-C, and C-O-H groups [26,27,28].

#### 3.4.4. XRD Analysis

The four main crystalline variants of cellulose are cellulose I, cellulose II, cellulose III, and cellulose IV. Under certain conditions, these variants can transform into each other. In the X-ray diffraction data, the positions of characteristic diffraction peaks of cellulose I are 2θ = 14.8° (10$\overset{-}{1}$), 16.6° (101), 22.7° (002), and 34.6° (040) [29]. From the characteristic diffraction peaks in Figure 6, the crystal structure of the fabrics before and after finishing did not change significantly and belonged to the cellulose I crystal. The results indicated that the crystal structure of the fabric was minimally affected by high-temperature baking under acidic conditions.

#### 3.4.5. TG Analysis

There are three stages during cellulose fiber pyrolysis: initial cracking (<370 °C), main cracking (370–430 °C), and residue cracking (>430 °C). The physical properties and weightlessness of fiber that takes place is interrelated with the amorphous region in the initial cracking; the weight loss rate of pyrolysis is fast, primarily in the fiber crystallization region, and the fiber is gradually decomposed to levoglucosan during main cracking, which is then degraded into various combustible gas products; residue cracking is mainly dehydration and carbonization reaction [30].

The thermal stability of the finished fabric (O-C) and the finished fabric (GPA-C) in the nitrogen environment can be observed in Figure 7, and we can know the thermal performance parameters of the fabric: initial decomposition temperature (Ti), maximum weight loss rate temperature (Tmax), extrapolation residue content (CR), and initial temperature (Te). See Table 3 for the detailed data.

Compared with the initial decomposition temperature of 256 °C of the finished fabric, the initial decomposition temperature of the finished fabric was reduced to 200 °C, which may be because the stability of the C-C structure is higher than that of the P-O-C structure [31]. During the combustion process, the phosphorus-containing flame retardant of PA preferentially decomposed the cotton fabric to generate phosphoric acid derivatives, and these decomposition products promoted the carbonization of the cotton fabric, thus reducing the initial decomposition temperature. The residue content increased (10%→48%) [32,33]. Table 2 shows that the Te and Tmax of the C-O fabric were above 300 °C, 345 °C, and 380 °C, respectively. After finishing, the Te and Tmax of the fabric decreased by about 30 °C, but the residue content increased to 48% at 700 °C. This may be because the phosphorous flame retardant on the fabric was decomposed to phosphoric acid at a high temperature, which polymerized into polyphosphoric acid or reacted with the -OH group at the C6 position. This caused the pyran ring to break and accelerate the degradation and dehydration of the cotton fabric under certain conditions, thus reducing Te and Tmax, promoting the formation of a coke layer, and greatly increasing the residue content after combustion [34]. Next, the coke layer covered the surface of the fiber, which isolated the fiber from the O_2_, suffocated the combustion, retarded the thermal decomposition reaction, thus, achieving the purpose of flame retardancy of the fabric [35].

#### 3.4.6. TG-IR Analysis

The TG-IR spectra before and after finishing are shown in Figure 8 to study changes in the gas produced during fiber pyrolysis. In general, due to the addition of the flame retardant, the composition and yield of complex small molecule gases pyrolyzed were different because of the reforming reactions and fracture of functional groups. Exceptionally light gases, such as H_2_O, CO, CO_2_, and CH_4_, a tremendous number of aldehydes, acids, phenols, and other substances were formed [36]. As shown in Figure 8, the amount of gas released during fiber pyrolysis was the most near 380 °C, which is consistent with the thermogravimetric analysis results. The CO_2_ asymmetric stretching vibration absorption peak at 2360 cm^−1^ was evidently strong. CO and CH_4_ were found at 2180 cm^−1^ or 2110 cm^−1^ and 2818 cm^−1^. Furthermore, the peaks near 1108 cm^−1^ and 1743 cm^−1^were attributed respectively to the stretching vibration of C-O and C=O, which were various aldehydes, ketones, and other macromolecular substances [21,37].

Compared with those of the untreated fabric, the infrared characteristic absorption peaks of the treated cotton fabric showed many significant changes during thermal decomposition. The absorption peak intensity of the H_2_O molecules enhanced at 3500–3800 cm^−1^ basically run through the entire pyrolysis process. This finding was mainly attributed to the production of phosphoric acid, which polymerized into polyphosphoric acid under strong heat. This acid was a dehydrant and absorbed a large amount of heat, thereby reducing the temperature and slowing down the thermal decomposition process. The characteristic peak intensities of CO_2_ at 2360 cm^−1^ and CO at 2180 or 2110 cm^−1^ weakened in Figure 8B compared with those in Figure 8A. The CO_2_ and CO gases released during pyrolysis mainly originated from the crack of C-O and C=O in fiber molecules, respectively. The coke layer formed during pyrolysis isolated the contact between the internal polymer and oxygen and inhibited the combustion of fiber [38,39].

## 4. Conclusions

Flame retardant and wrinkle resistance fabrics were obtained by vinyl modification of phytic acid and itaconic acid through in situ polymerization crosslinking finishing. The antiwrinkle and thermal properties of the finished fabrics were greatly improved. During the pyrolysis process, the maximum weight loss rate temperature decreased, water release increased, and the carbon residue content increased after combustion, which effectively suppressed or slowed down the burning degree of the textile, providing a way of thinking for the flame-retardant antiwrinkle finishing of the fabric.

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
