# Peer review of "Vinylated Modification of Biophytic Acid and Flame-Retardant/Crease-Proofing Finishing of Cotton Fabrics via In Situ Copolymerization"

_materials, 2022, doi:10.3390/ma16010286_

Round 1

Reviewer 1 Report

The authors of the manuscript titled: "Flame-retardant and Crease-proofing Finishing of Cotton Fabrics via In-situ Copolymerization of GPA/IA" present their work at designing a flame resistant fabric combining cotton with phytic acid. While the experimental work seems to be in order, the quality of its presentation needs to be seriously improved. The greatest flaw of the manuscript is the English grammar, which is rather poor. The article needs to be thoroughly proofread to fix all the broken sentences and misused expressions.

There are also issues with multiple figures:
The NMR results (equation 1) need to be presented in a better comprehensible manner. 
Scheme 1 formatting seems to be broken (parts of the image are overlapping).
Figure 2 caption misses (B) marking.
Figure 4 caption is very insufficient. The sections of the image need to be marked and described properly in the caption.
Figure 5 caption needs to describe the section of the figure according to its (A-D) markings.
Figure 7 caption misses the (b) and (b*) markings.

The abstract needs to be slightly reworked. The description of the exact ratios of the sample components is superfluous, some of the abbreviations are not explained.

Some parts of the experimental section are confusing - there are many different units of concentration used - g/L, mol/L, ppm, weight %, mass ratio. This is rather disorienting.
The meaning of the sentence: "In brief, 0.02 mol 70 % phytic acid solution (PA) and 80 mL of acetone were added into 250 mL beaker." is unclear. Perhaps there is mistake in the units?

I believe these issues are quite minor, yet there is a lot of them. If these are resolved, the manuscript should be eligible for publication.

Author Response

1、While the experimental work seems to be in order, the quality of its presentation needs to be seriously improved. The greatest flaw of the manuscript is the English grammar, which is rather poor. The article needs to be thoroughly proofread to fix all the broken sentences and misused expressions.

Response: The manuscript had been revised about the English.

2、The NMR results (equation 1) need to be presented in a better comprehensible manner.

Response: Thank you for your proposal. The NMR results had been presented via table in the manuscript.  

3、Scheme 1 formatting seems to be broken (parts of the image are overlapping).

Response: the scheme 1 had been revised in the manuscript.  

4、Figure 2 caption misses (B) marking.

Response: I'm sorry, the authors were careless and the text has been revised.

5、Figure 4 caption is very insufficient. The sections of the image need to be marked and described properly in the caption. Figure 5 caption needs to describe the section of the figure according to its (A-D) markings. Figure 7 caption misses the (b) and (b*) markings.

Response: The sections of the image had been marked and described properly in Figure 4, 5 and 7 captions in the manuscript.

6、The abstract needs to be slightly reworked. The description of the exact ratios of the sample components is superfluous, some of the abbreviations are not explained.

Response: The abstract had been revised in the article, as follows:

The vinyl phytic acid (GPA) was prepared by bio phytic acid (PA) and glycidyl methacrylate (GMA),which double bonds are introduced into the phytic acid molecule to increase the active groups in the phytic acid molecule. Furthermore, itaconic acid (IA) containing two unsaturated double bonds and GPA were in-situ polymerized and crosslinked on the surface of cotton fabrics, that the flame-retardant and crease-proofed fabrics were obtained. the effects of GPA, IA, and initiator on the flame-retardant and crease-proofing properties of the fabrics were analyzed by single-factor and double-dip-double-nip experiment. The Flame retardant and wrinkle resistant fabric was obtained which the limiting oxygen index (LOI) and wrinkle recovery angle (WRA) were 28 % and 270°, respectively. During combustion, the thermal properties of fabrics were changed, typically, the extrapolated initial temperature (Te) decreased, and the moisture release increased. The fabrics after burning had good shape retention, and the carbon residue content increased to 48 %, which effectively inhibited or slowed down the combustion and heat release of the textiles. However, the whiteness, mechanical properties, and washability of the products need to be further improved.

7、Some parts of the experimental section are confusing - there are many different units of concentration used - g/L, mol/L, ppm, weight %, mass ratio. This is rather disorienting.

Response: Many different units of concentration and related sentences had been revied in the manuscript.

Reviewer 2 Report

1-     The state-of-the-art should be mentioned clearly in introduction.

2-     It is suggested that the authors should discuss the following articles:

https://www.sciencedirect.com/science/article/abs/pii/S0254058418307235

https://link.springer.com/article/10.1007/s11998-022-00633-x

3-     More information about the figures should be added to the caption of Figures 4 and 5. The SEM images should be labeled. 

4-     Obtained results must be compared with other related literature.

5-     What is the effect of flame retardant finishing on breathability and conformability of fabric?

6-     Please explain about the washing and rubbing fastness of finished fabric.

Author Response

1、The state-of-the-art should be mentioned clearly in introduction. It is suggested that the authors should discuss the following articles:

https://www.sciencedirect.com/science/article/abs/pii/S0254058418307235  

https://lin k.springer.com/article/10.1007/s11998-022-00633-x

Response: The state-of-the-art and the first article provided by reviewer had been supplemented in the introduction, however, the second article provided by reviewer couldn`t opened unfortunately.

Therefore, it has become a new development direction of flame retardant for cotton to seek new environmental flame retardant with high efficiency, excellent durability, low smoke, low toxicity or no-toxicity, no formaldehyde. Some scholars had prepared a new type of dendriform poly (amine) dendrimer flame retardant, which was used for flame retardant finishing of cotton fabric under the action of citric acid. The limiting oxygen index exceeds 25 %. The carbonization length of the treated sample after five times of washing was slightly longer than that of the unwashed sample [3].

Furthermore, researchers of Southwest University [14] used urea to modify phytic acid to prepare ammonium phytate, and then used dicyandiamide as catalyst to finish cotton fabric. The LOI of the flame retardant fabric was 43.2 %, and after 30 times of washing, the LOI was 30.5 %. Compared with the untreated cotton fabric, the peak heat release rate and total heat release rate of the treated cotton fabric decreased significantly. Subsequently, based on ammonium phytate, the researchers [15] synthesized raffinose cross-linked boric acid-ammonium phytate (r-GBAP) flame retardant by one pot method to enhance the durable flame retardancy of cotton fabrics. When the amount of flame retardant (r-GBAP) was 100 g/L, the LOI of the flame retardant fabric was 41.6% which has water washing resistance. In addition, Li [16] selected ammonium phytate and chitosan as environment-friendly flame retardants to improve the flame retardancy and antibacterial properties of viscose fabrics. The thermal stability of flame-retardant viscose fabric in the higher temperature zone was significantly improved. The limiting oxygen index of the viscose layer by layer assembly of two layers was 29 %, and 39.0 % coke residue was retained at 700 ° C. however, the flame retardant and crease resistant finishing of cotton fabric with phytic acid has not been reported.

2、More information about the figures should be added to the caption of Figures 4 and 5. The SEM images should be labeled.

Response: The sections of the image had been marked and described properly in Figure 4, 5 in the manuscript.

3、Obtained results must be compared with other related literature.

Response: Thank you for your good proposal. However, the ethylene modification of phytic acid for flame retardant and wrinkle resistant finishing of cotton fabrics has not been reported so far, therefore, there is no comparative analysis with the literature. Although the flame retardant and wrinkle resistant effect of the preliminary study needs to be improved, this method is feasible. The follow-up research focuses on the modified phytic acid finishing method of fabrics to enhanced the washability.

4、What is the effect of flame retardant finishing on breathability and conformability of fabric?

Response: Sorry. In situ polymerization and crosslinking method with vinyl phytic acid has a great impact on the whiteness and strength of the fabric. In this case, it is regrettable that no other wearing properties of the fabric, such as air permeability and comfort, have been studied. In the method of subsequent improvement, a detailed study on the wearing properties of the finished fabric will be carried out.

5、Please explain about the washing and rubbing fastness of finished fabric.

Response: the washing fastness needs to be further improved. The LOI of the finishing fabric reached 28.6, and the WRA was increased to 265 °; both parameters decreased after washing because a number of IA was not reacted with the hydroxyl group of the fabric. In the presence of the catalyst, the hydroxyl group in the molecular structure of PA blocked the grafting reaction of IA and the fabric. The steric hindrance of PA also reduced the polymerization with IA. Therefore, the LOI and WRA of the fabrics decreased to a certain extent. We generally test the rubbing fastness of dyed fabrics; however, the rubbing fastness of finished fabrics is not evaluated unfortunately.

Reviewer 3 Report

What about its effect on dyed cotton fabric? It should be added if possible

Author Response

1、What about its effect on dyed cotton fabric? It should be added if possible。

Response: Thank you for your proposal. Unfortunately, studies found that in situ polymerization and crosslinking method with vinyl phytic acid has a great impact on the whiteness and strength of the fabric, Therefore, it has a great influence on dyed fabric. We will do further research after the method is improved. thank you.

Reviewer 4 Report

The authors have developed FR and anticrease treatment for cellulose using phytic acid. The work presented here is of poor scientific quality with unnecessary data. the work presented has no meaning as the durability of such treatment is not evaluated. the proof of cellulose modification cannot be confirmed via any of the experimental data. Not sure why Xray and XPS data are presented. The synthesis data especially the NMR data is not clear. No spectra is discussed. The manuscript doesn't provide any new information which is not already known in the literature.

Author Response

The authors have developed FR and anticrease treatment for cellulose using phytic acid. The work presented here is of poor scientific quality with unnecessary data. the work presented has no meaning as the durability of such treatment is not evaluated. the proof of cellulose modification cannot be confirmed via any of the experimental data. Not sure why Xray and XPS data are presented. The synthesis data especially the NMR data is not clear. No spectra is discussed. The manuscript doesn't provide any new information which is not already known in the literature.

Response: thank you. the ethylene modification of phytic acid for flame retardant and wrinkle resistant finishing of cotton fabrics has not been reported so far, therefore, authors hope to provide a method. Although the flame retardant and wrinkle resistant effect of the preliminary study needs to be improved, we discussed the washability of the finished fabric in the manuscript (3.4.1 Basic properties of finished fabrics). NMR data had been revied in the article. Xray mainly was used to discuss the influence on the crystal area of the original and finished fabric. authors will focus on finishing method of fabrics to enhanced the washability based on ethylene phytic acid in the follow-up research.

Round 2

Reviewer 2 Report

The authors have satisfactorily addressed all the reviewer's concerns. Hence, I recommend the publication of the manuscript.